# Optimal opportunistic screening of atrial fibrillation using pulse palpation in cardiology outpatient clinics: Who and how

**Jordi Bañeras**[1,2,3]*, **Ivana Pariggiano**[4,5], **Eduard Ródenas-Alesina**[1], **Gerard Oristrell**[1], **Roxana Escalona**[1], **Berta Miranda**[1], **Pau Rello**[1], **Toni Soriano**[1], **Blanca Gordon**[1], **Yassin Belahnech**[1], **Paolo Calabrò**[4,5], **David García-Dorado**[1,2,3†], **Ignacio Ferreira-González**[1,2,6], **Joaquim Radua**[7,8,9,10]

**1** Department of Cardiology, Hospital Universitari Vall d'Hebron, Barcelona, Spain, **2** Vall d'Hebron Research Institute, Vall d'Hebron Hospital, Autonomous University of Barcelona, Barcelona, Spain, **3** CIBER CV, ISC-III, Madrid, Spain, **4** Division of Clinical Cardiology, A.O.R.N. "Sant'Anna e San Sebastiano", Caserta, Italy, **5** Department of Translational Medical Sciences, University of Campania "Luigi Vanvitelli", Naples, Italy, **6** CIBER ESP, ISC-III, Madrid, Spain, **7** Institut d'Investigacions Biomèdiques August Pi i Sunyer (IDIBAPS), Barcelona, Spain, **8** CIBERSAM, ISC-III, Madrid, Spain, **9** King's College London, London, United Kingdom, **10** Karolinska Institutet, Stockholm, Sweden

† Deceased.
* jbaneras@vhebron.net

**Data Availability Statement:** Data from this study are available upon request. In that case the data used in this study contain potentially sensitive information. Requests may be sent to the Ethics

## Abstract

### Background

Atrial fibrillation (AF) remain a prevalent undiagnosed condition frequently encountered in primary care.

### Objective

We aimed to find the parameters that optimize the diagnostic accuracy of pulse palpation to detect AF. We also aimed to create a simple algorithm for selecting which individuals would benefit from pulse palpation and, if positive, receive an ECG to detect AF.

### Methods

Nurses from four Cardiology outpatient clinics palpated 7,844 pulses according to a randomized list of arterial territories and durations of measure and immediately followed by a 12-lead ECG, which we used as the reference standard. We calculated the sensitivity and specificity of the palpation parameters. We also assessed whether diagnostic accuracy depended on the nurse's experience or on a list of clinical factors of the patients. With this information, we estimated the positive predictive values and false omission rates according to very few clinical factors readily available in primary care (age, sex, and diagnosis of heart failure) and used them to create the algorithm.

Committee of Hospital Universitari Vall d'Hebron (Contact phone: +34934894010, email: CEIC@vhir. org".

**Funding:** JR is supported by a Miguel Servet Research Contract (CPII19/00009) from the Instituto de Salud Carlos III and co-funded by European Union (ERDF/ESF, 'Investing in your future'). The funders had no role in study design, data collection and analysis, decision to publish, or preparation of the manuscript.

**Competing interests:** The authors have declared that no competing interests exist.

## Results

The parameters associated with the highest diagnostic accuracy were palpation of the radial artery and classifying as irregular those palpations in which the nurse was uncertain about pulse regularity or unable to palpate pulse (sensitivity = 79%; specificity = 86%). Specificity decreased with age. Neither the nurse's experience nor any investigated clinical factor influenced diagnostic accuracy. We provide the algorithm to select the ≥40 years old individuals that would benefit from a pulse palpation screening: a) do nothing in <60 years old individuals without heart failure; b) do ECG in ≥70 years old individuals with heart failure; c) do radial pulse palpation in the remaining individuals and do ECG if the pulse is irregular or you are uncertain about its regularity or unable to palpate it.

## Conclusions

Opportunistic screening for AF using optimal pulse palpation in candidate individuals according to a simple algorithm may have high effectiveness in detecting AF in primary care.

## Introduction

Atrial fibrillation (AF) is the most common arrhythmia [1], and its prevalence rises with age, from about 2% in the whole population to about 10%-17% in individuals aged 80 years or older [2]. It carries a near five-fold risk of ischemic stroke [3], especially severe stroke [4], and a near two-fold mortality risk [5]. Fortunately, it can be diagnosed using a simple, low-cost test, an electrocardiography (ECG), and anticoagulation can reduce the risk of stroke by approximately 60% and mortality by almost a quarter [6].

Relevantly, this arrhythmia is asymptomatic in about one-third of the patients [7], but the morbidity and mortality rates of silent AF are similar to those of symptomatic AF [8]. The absence of symptoms and the existence of an effective stroke prevention treatment call for the creation of effective screening programs.

Two classic population active screening strategies include opportunistic and systematic screening [9], and both have been shown to detect additional AF cases over current practice [10]. Indeed, the current Guidelines for the Primary Prevention of Stroke from the American Heart / Stroke Association (AHA/ASA) recommend active screening for AF in the primary care setting in individuals >65 years of age by pulse palpation followed by ECG if palpation is positive for irregular pulse [11]. Similarly, the latest European Society of Cardiology (ESC) Guidelines for the management of AF recommend opportunistic screening for AF by pulse taking or ECG rhythm strip in individuals >65 years of age [12].

However, some voices have raised criticism about the current AF screenings initiatives. For example, the US Preventive Services Task Force concludes that the current evidence is insufficient to assess the balance of benefits and harms of screening for AF with an ECG [13]. In addition, we neither know the optimal pulse palpation parameters (e.g., arterial territory or duration of the measure) nor how to interpret palpations in which the professional is uncertain about pulse regularity or unable to palpate a pulse.

In this context, we investigated for the first time which pulse palpation parameters are more accurate to detect AF. We also estimated the predictive values to create an easy algorithm so that the primary care physician readily knows which individuals may benefit from this

screening and which may not. Our first aim was to establish which palpation parameters would optimize the sensitivity and specificity of palpation. The second aim was to check whether the sensitivity or specificity of palpation might depend on a range of clinical factors. This checking was necessary for two reasons. On the one hand, we had to ensure that sensitivity and specificity did not depend on heart conditions. Thus, we could safely translate our results to the general population. On the other hand, we had to know the sensitivity and specificity of each algorithm stratum that we could create later. Finally, the final aim was to estimate the positive predictive values (PPV) and false omission rates (FOR) separately for strata of age, sex, and the presence of very few readily available clinical factors. We hope that physicians might successfully apply this algorithm to select individuals in whom to optimally palpate the pulse, efficiently improving the detection of AF and thus the prevention of stroke and its costs.

## Materials and methods

We conducted the study in accordance with the ethical principles of the Declaration of Helsinki and Good Clinical Practice, and the Ethics Committee of Hospital Vall d'Hebron approved its protocol (PR(AG)151/2013). All participants received information on the study, and verbal informed consent was obtained before enrolment.

## Participants

We invited all adults attending any of the four Cardiology outpatient clinics of the northern region of Barcelona (Horta, Sant Andreu, Xafarines, and Rio de Janeiro), Spain, from 1st June 2014 to 31st July 2016 to participate in this study. These centers provide health care to 403,000 habitants (24.4% of the population of Barcelona). A nurse routinely performs an ECG that a cardiologist later reports in each visit.

After checking the report of the ECG, we excluded: a) patients with pacemakers, because when pacing, the rhythm is regular and it is not possible to detect AF with pulse palpation; b) patients with atrial flutter, because sometimes the rhythm is regular; and c) patients in whom an ECG was not performed or could not be verified.

## Pulse palpations

To ensure blindness, the nurse palpated the pulse before performing the routine ECG and before asking for any information about the patient's pathology. According to a randomization sheet, the palpation was of a specified arterial territory (right radial, right carotid, or both) during a specified time (10 seconds, defined by nursing consensus, versus the necessary time). With the strategy of preset time vs necessary time, we wanted to incorporate the variable of whether looking at a clock (10 seconds) had an effect on the perception of regularity due to synchronization phenomena and decreased the effectiveness of detecting AF. The outcome was coded as regular, irregular, "uncertain about pulse regularity", or "unable to palpate a pulse". When palpation was specified to be of both radial and carotid territories, and they differed, the outcome had to be coded as irregular if one territory was palpated irregular. If the nurse could not palpate pulse in one territory, the outcome had to be based on the other territory. Immediately after pulse palpation, the nurse conducted a 12-lead ECG, considered the gold standard for detecting AF after reading by a cardiologist [14], and which we used as the reference standard.

Before the study, the nurses had received 30 minutes of training by a cardiologist in the clinical assessment of the pulse rhythm (e.g., to neither press too much nor massage the carotid artery).

## Statistical analysis

We estimated the sensitivity of pulse palpation to detect AF as the proportion of individuals with AF that had positive palpation (i.e., an irregular pulse). Similarly, we estimated the specificity of pulse palpation to detect the absence of AF as the proportion of individuals without AF with negative palpation (i.e., a regular pulse). We estimated this proportion with a logistic linear mixed-effects model that included the nurse as a random factor in both cases. Finally, we calculated the balanced diagnostic accuracy as the average of sensitivity and specificity.

To establish which palpation parameters would optimize the sensitivity and specificity, we repeated the analyses, including the following independent variables in the logistic linear mixed-effects model: the arterial territory palpated (right radial, right carotid, or both), the duration of palpation (10 seconds versus the necessary time), and the experience of the nurse (two groups defined by the median number of pulse palpations conducted by the nurse within the study before the current palpation). In addition, we considered four possible definitions of positive palpation: a) only when the nurse coded the pulse as irregular; b) when the nurse coded the pulse as irregular or was uncertain about its regularity; c) when the nurse coded the pulse as irregular or was unable to palpate it; d) when the nurse coded the pulse as irregular, was uncertain about its regularity, or was unable to palpate it. Finally, we selected the palpation parameters that yielded the highest balanced diagnostic accuracy.

To check whether the sensitivity or specificity of palpation might depend on clinical factors, we repeated the analyses, including the following independent variables in the logistic linear mixed-effects model: age (<65 vs. ≥65 year old), sex, presence or absence of high blood pressure, dyslipidemia, diabetes mellitus, peripheral artery disease, hypertensive heart disease, ischemic cardiomyopathy, right bundle branch block, left bundle branch block, lower or higher heart rate (as defined by its median), lower or higher ejection fraction (as defined by its median), lower or higher body mass index (as defined by its median), and permanent or paroxysmal AF.

Finally, we estimated the PPV as the proportion of individuals with positive palpation that really had AF and the FOR as the proportion of individuals with negative palpation that indeed had AF. To estimate PPV and FOR we needed the sensitivity and specificity of palpation (which we had from the previous steps) and the prevalence of AF in the specific stratum. We obtained the latter from a study of the prevalence of AF in Spain [15]. We considered the four clinical factors that influenced the prevalence with p≤0.001 in the multivariate model of that study: age, sex, central obesity, and heart failure.

We conducted all analysis with the R packages "lme4" and "multcomp" [16, 17]. To correct for the 40 assessments (4 definitions of positive palpation × 5 comparisons of palpation parameters × 2 measures–sensitivity and specificity–), we established the level of statistical significance at 0.001 (i.e., ~0.05 / 40).

## Creation of the algorithm

We used the estimations of PPV and FOR in each stratum to create a simple algorithm to help primary care physicians know which individuals may benefit from a palpation screening. We deemed that the screening would be unnecessary in individuals with PPV<10% because PPV<10% means that they would have a low probability of AF (<10%) even if palpation were positive. Similarly, we considered recommending ECG without the need for prior palpation in individuals with FOR>10% because FOR>10% means that they would have a high probability of AF (>10%) even if palpation were negative. We deemed it appropriate for the remaining individuals to conduct pulse palpation, followed by an ECG when positive.

It is important to remark that this cut-off point (10%) is only a suggestion, and other cut-off points may be more appropriate in a specific health system. Fortunately, the algorithm may be easily recreated with other cut-off points.

## Results

The ten nurses assessed 7,844 pulses. After discarding pacemakers, atrial flutters, and ECGs not performed or verified (Fig 1), we could include 7,430 from 5,508 patients (mean age ±SD = 70±13 years, 43% females, see Table 1 for further details). The least frequent pulse palpations per nurse were 36 and the next 210, and the most frequent palpations per nurse were 1696 and 2560. The palpated arterial territory was carotid in 33.3% palpations, radial in 33.7% palpations, and both carotid and radial in 33.1% palpations. The duration of palpation was 10 seconds in 49.9% palpations, and the necessary time in 50.1%. The nurses coded 63.7% palpations as regular, 23.3% irregular, 6.1% "uncertain about pulse regularity", and 6.9% "unable to palpate a pulse". No cardiologists reported doubts in the electrocardiographic detection of AF. The ECG showed that 81.6% of pulses were in sinus rhythm and 18.4% in AF.

We observed the highest balanced diagnostic accuracy when we considered that palpation was positive when it had been coded as irregular, "uncertain about pulse regularity", or "unable to palpate pulse". Using this definition of positive palpation, the specificity was maximum for radial palpations (p<0.001, Table 2). Specifically, radial palpations showed 79% sensitivity, 86% specificity, and 83% balanced diagnostic accuracy. Using the same definition, the duration of palpation and the nurse's experience did not show statistically significant effects on specificity, and no palpation parameters showed statistically significant effects on sensitivity.

Specificity was statistically significantly lower in older individuals (91% in <65 years and 84% in ≥65 years, p<0.001). We did not detect statistically significant influences of any other clinical factors.

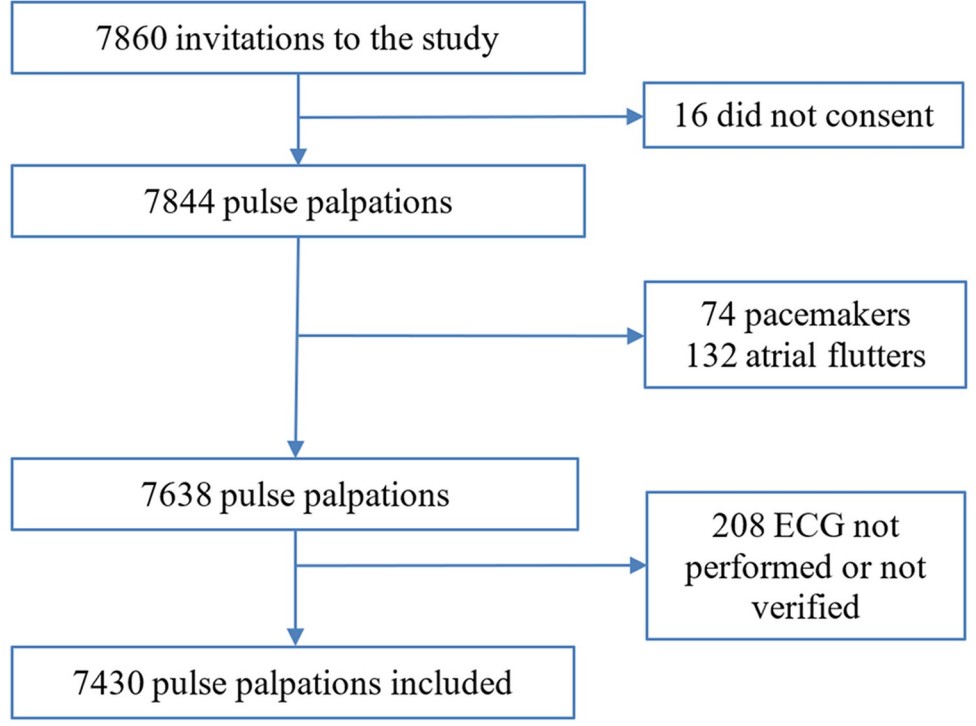

**Fig 1. Study flow chart.**

**Table 1. Description of the sample.**

| | Individuals | | | | Palpations | | |
|---|---|---|---|---|---|---|---|
| | **All** | **Sinusoidal rhythm** | **Atrial fibrillation** | **Both[a]** | **All** | **Sinusoidal rhythm** | **Atrial fibrillation** |
| Sample size | 5,508 | 4,490 | 922 | 96 | 7,430 | 6,060 | 1,370 |
| Age (mean±SD) | 70±13 | 69±14 | 77±8 | 77±9 | 71±13 | 69±13 | 77±8 |
| Females (%) | 43% | 42% | 48% | 43% | 42% | 41% | 47% |
| Body mass index (mean±SD) | 29±5 | 29±5 | 30±5 | 29±5 | 29±5 | 29±5 | 30±5 |
| Heart rate (mean±SD) | 69±13 | 68±11 | 75±16 | 71±14 | 69±13 | 68±12 | 74±18 |
| Ejection fraction (mean±SD) | 57±10 | 57±10 | 56±10 | 57±10 | 57±10 | 57±10 | 56±10 |
| High blood pressure (%) | 74% | 72% | 83% | 85% | 74% | 72% | 84% |
| Dyslipidemia (%) | 57% | 58% | 51% | 64% | 58% | 59% | 53% |
| Diabetes mellitus (%) | 26% | 25% | 26% | 36% | 27% | 26% | 28% |
| Peripheral artery disease (%) | 7% | 7% | 5% | 11% | 8% | 8% | 8% |
| Hypertensive heart disease (%) | 34% | 32% | 44% | 33% | 34% | 32% | 44% |
| Ischemic cardiomyopathy (%) | 40% | 43% | 25% | 39% | 43% | 46% | 29% |
| Right bundle branch block (%) | 8% | 8% | 6% | 8% | 8% | 8% | 6% |
| Left bundle branch block (%) | 5% | 5% | 5% | 5% | 5% | 5% | 6% |

(a) Patients whose ECG showed sinusoidal rhythm in one visit and atrial fibrillation in another.

Tables 3 and 4 show the creation and a summary of the clinical algorithm to indicate pulse palpation (and if positive ECG) or ECG (without previous pulse palpation) in primary care.

## Discussion

Despite pulse palpation being widely recommended to screen for AF, there is little evidence about the optimal palpation parameters or its PPV and FOR in general practice settings. To shed light on this field, we report the results of a large study to assess the diagnostic accuracy (sensitivity and specificity) of pulse palpation as a screening tool to detect AF depending on palpation parameters and clinical factors. To include many individuals with AF, we conducted the study in cardiology outpatients, in whom the prevalence of AF is higher than in the general

**Table 2. Diagnostic accuracy of pulse palpation for detecting atrial fibrillation depending on palpation parameters.**

| Palpation parameters | | Diagnostic accuracy for detecting atrial fibrillation | | |
|---|---|---|---|---|
| **Definition of positive palpation** | **Arterial territory** | **Sensitivity** | **Specificity** | **Balanced diagnostic accuracy** |
| Irregular | Carotid | 46% | 92% | 69% |
| | Radial | 72% | 90% | 81% |
| | Carotid + Radial | 66% | 87% | 77% |
| Irregular or "uncertain about pulse regularity" | Carotid | 65% | 80% | 73% |
| | Radial | 77% | 88% | 82% |
| | Carotid + Radial | 73% | 81% | 77% |
| Irregular or "unable to palpate a pulse" | Carotid | 63% | 82% | 73% |
| | Radial | 74% | 88% | 81% |
| | Carotid + Radial | 69% | 85% | 77% |
| Irregular, "uncertain about pulse regularity", or "unable to palpate a pulse" | Carotid | 82% | 69% | 75% |
| | Radial | 79% | 86% | 83% |
| | Carotid + Radial | 76% | 79% | 78% |

The reference standard was established by 12-lead electrocardiography recorded immediately after pulse palpation.

**Table 3. Predictive positive values (PPV) and false omission rates (FOR) of pulse palpation to detect atrial fibrillation depending on clinical factors used to create the algorithm to indicate pulse palpation (and if positive ECG) or ECG (without previous pulse palpation) in primary care.**

| Age group | Age | Heart failure | Central obesity | PPV | | FOR | |
|---|---|---|---|---|---|---|---|
| | | | | Males | Females | Males | Females |
| 40–59 | 40–44 | no | no | 4% | 1% | 0% | 0% |
| | | | yes | 6% | 2% | 0% | 0% |
| | | yes | no | 22% | 9% [a] | 1% | 0% |
| | | | yes | 32% | 15% | 1% | 0% |
| | 45–49 | no | no | 3% | 1% | 0% | 0% |
| | | | yes | 5% | 2% | 0% | 0% |
| | | yes | no | 21% | 9% [a] | 1% | 0% |
| | | | yes | 31% | 14% | 1% | 0% |
| | 50–54 | no | no | 6% | 6% | 0% | 0% |
| | | | yes | 9% | 10% [a] | 0% | 0% |
| | | yes | no | 32% | 35% | 1% | 1% |
| | | | yes | 45% | 47% | 2% | 2% |
| | 55–59 | no | no | 4% | 5% | 0% | 0% |
| | | | yes | 7% | 9% | 0% | 0% |
| | | yes | no | 25% | 30% | 1% | 1% |
| | | | yes | 36% | 42% | 1% | 2% |
| 60–69 | 60–64 | no | no | 21% | 18% | 1% | 1% |
| | | | yes | 31% | 27% | 1% | 1% |
| | | yes | no | 68% | 62% | 5% | 4% |
| | | | yes | 78% | 74% | 8% | 7% |
| | 65–69 | no | no | 13% | 9% [a] | 1% | 0% |
| | | | yes | 20% | 14% | 1% | 1% |
| | | yes | no | 54% | 43% | 5% | 4% |
| | | | yes | 66% | 56% | 9% | 6% |
| ≥70 | 70–74 | no | no | 28% | 15% | 2% | 1% |
| | | | yes | 40% | 23% | 3% | 1% |
| | | yes | no | 75% | 58% | 13% | 6% [a] |
| | | | yes | 83% | 70% | 20% | 10% |
| | 75–79 | no | no | 22% | 12% | 1% | 1% |
| | | | yes | 32% | 19% | 2% | 1% |
| | | yes | no | 69% | 51% | 10% | 5% [a] |
| | | | yes | 79% | 64% | 15% | 8% [a] |
| | ≥80 | no | no | 32% | 31% | 2% | 2% |
| | | | yes | 44% | 43% | 4% | 4% |
| | | yes | no | 78% | 78% | 15% | 15% |
| | | | yes | 86% | 86% | 23% | 22% |

We recommended doing nothing if both PPV and FOR are <10% (both in green), pulse palpation if PPV is >10% and FOR <10% (PPV in red and FOR in green), and electrocardiography (ECG) without previous pulse palpation if both PPV and FOR are >10% (both in red).

(a) For a few cells, we considered numbers <10% as >10% or vice-versa to match the global pattern.

population. As we detailed earlier, we checked that sensitivity and specificity did not depend on our sample's range of heart conditions. Thus, our results should be generalizable to the general population. However, we must note that if we had conducted this study in a sample from the general population, the number of individuals with AF would have been substantially

**Table 4. Practical summary of the algorithm to indicate pulse palpation (and if positive ECG) or ECG (without previous pulse palpation) to detect atrial fibrillation in primary care.**

| Screening for atrial fibrillation in ≥ 40 years old individuals | |
| --- | --- |
| Age | Recommendation |
| <60 years | If heart failure: palpation |
| | Otherwise: nothing |
| 60–69 years | Palpation |
| ≥70 years | If heart failure: ECG |
| | Otherwise: palpation |

If palpation is irregular, or you are uncertain about pulse regularity or unable to palpate pulse: do ECG.

smaller. Thus, the estimates of sensitivity would be imprecise. That said, we encourage a future study in general primary care patients to ensure the transferability of the results of this study.

Wee estimated the PPV and the FOR, i.e., the probabilities of having AF if pulse palpation is positive or negative in individuals with different clinical characteristics. PPV and FOR allowed us to create a simple clinical algorithm based on very few clinical factors to help know which individuals may benefit from a palpation screening in primary care. For example, there are few reasons to conduct the screening in individuals who will be very unlikely to have AF even if pulse palpation is positive or in individuals who will still have a significant probability of having AF even if palpation is negative. Conversely, there are clear reasons to conduct the screening in individuals with a significant probability of having AF if palpation is positive and a negligible probability if palpation is negative.

AF is common in the elderly and individuals with some clinical risk factors such as heart failure. In addition, its prevalence is increasing in western societies due to population aging and the increase of precipitating cardiovascular diseases. Screening these groups should identify many individuals with AF [2, 9], of which a critical proportion may be candidates to anticoagulant treatment to prevent stroke. For example, a previous systematic review found that single-time-point ECG recording detected unknown AF in 1.4% ≥65-year- individuals [18]. And the screening is effective because, in previous studies, more than 90% of participants with previously undiagnosed AF accepted initiation of oral anticoagulant treatment [19].

The gold standard for detecting AF is the 12-lead ECG [14], but its efficiency is hampered by its cost on human resources (it takes time). A straightforward strategy to improve the efficiency of AF detection consists of first selecting those individuals with a higher risk of AF, and the cheapest and quickest method to evaluate this risk is palpation of the pulse. Indeed, systematic pulse assessment during routine clinic visits followed by 12-lead ECG in those with an irregular pulse has been already reported to substantially increase the detection of AF [10].

In general, screening tests should have a very high sensitivity to detect all patients, even at the cost of a poorer specificity, because subsequent tests should discard the false positives. *Unfortunately, our sensitivity was only moderately high* (79%), while other studies have reported higher sensitivities [20]. We cannot rule out whether the true sensitivity might be thus higher, and a higher sensitivity would mean higher PPV and lower FOR and hence better prediction values. On the other hand, we found a relatively high specificity (86%), in this case, higher than previously described [14, 20]. In any case, our estimates may be not directly comparable to those of previous studies because most of the latter did not clearly report how they included or coded "uncertain about pulse regularity" and "unable to palpate pulse" palpations [10]. Morgan et al. also found high specificity (98%) but low sensitivity (54%) when considering an abnormal pulse as a 'continuous irregularity' [21]. Our results were closer to the

Morgan et al. definition of an abnormal pulse as a 'frequent or continuous irregularity' (sensitivity 72%, specificity 94%) [21]. Sudlow et al. reported a sensitivity over 90% but 71% specificity when any pulse that was not "regular" was considered abnormal [22].

The importance of age in AF may have led many screening studies to use a threshold of $\geq 65$ years [21, 22]. However, the prevalence of AF is not negligible in middle-aged people [23]. In this group, the annual incidence of stroke is 1.3% [24]. Younger people can also receive anticoagulation if they meet the CHA2DS2-VASc score criteria because 1–2 points are age>65–75 years, but the other 7 points are not age-related [12]. However, the low prevalence of comorbidities included in the CHA2DS2-VASc score amongst younger individuals may mean that many may not benefit from AF screening.

As in prior screening studies for AF, we used single time point measures, which are correct to detect permanent arrhythmia but may miss paroxysmal arrhythmia. Previous studies using intermittent ECGs substantially increased the detection of AF [25]. A recent study found that the detection increased with the number of screenings [26]. Thus, a repeated screening of the individuals using opportunistic pulse palpation during GP consultations is likely cost-effective.

In the recent years, there have appeared new AF detection methods. These can be divided into record ECG tracings (single or multilead, in intermittent or continuous format, of various durations) and use non-ECG techniques such as pulse photoplethysmography, providing each method different diagnostic accuracy. These technologies are under examination to address the challenge of screening for AF in the community, including iPhone ECG devices by pharmacists, general practice nurses, and receptionists [27]. Their reported sensitivities and specificities are acceptable [28]. However, given the lack of external validation in most studies, we should be careful about their general use until more evidence is available, and their effectiveness must still be evaluated [9, 13]. In a recent smartwatch study that included 419,297 participants, the probability of receiving an irregular pulse notification was low, and among those who received a notification of an irregular pulse, only 34% had AF on subsequent ECG patch reading [29], so the impact on cost-benefit should be studied.

In addition to the costs, when an active AF screening program is done, only a proportion of the individuals agree to participate (e.g., 54% in the STROKESTOP study) [19]. Conversely, our proposed screening would consist of simple and inexpensive pulse palpation, which does not raise anxiety and is acceptable to most individuals [10]. Indeed, in our study, only 16 individuals did not consent. With the algorithm we propose, the general physician will have a clear indication of whether it is necessary or not to perform an ECG in less than a minute. This short screening duration is of utmost importance, given that general practice consultations often last 5–10 minutes.

Finally, regarding to treatment, we do not know whether silent AF episodes require anticoagulation therapy, since no trials have assessed the benefits and harms of anticoagulation treatment among screen-detected populations, despite the fact that three quarters of the detected patients end up receiving anticoagulation [30]. In this context, the United States Preventive Services Task Force concludes that the current evidence is insufficient to assess the balance of benefits and harms of screening for AF [31]. However, in the ASSERT trial subclinical AF was associated with an increased risk of clinical AF (HR 5.56; 95% CI 3.78–8.17; P < 0.001) and of ischaemic stroke or systemic embolism (HR 2.49; 95% CI 1.28–4.85; P = 0.007). In the next years, several trials should be able to tell us if anticoagulation therapy should be used for short-lasting episodes diagnosed on cardiac devices [32].

There are several limitations to our study. First, we excluded patients with pacemakers or atrial flutters. While individuals know in advance that they have a pacemaker, this is not true for atrial flutters. However, this limitation should be minor because the incidence of atrial flutter is substantially lower than AF. Second, we did not investigate the accuracy of more recently

developed single-lead ECG devices. Future studies might investigate the effectiveness of these devices in detecting unknown AF in primary care. Third, we created the algorithm based on a study of the prevalence of atrial fibrillation in Spain [15]. For countries where the prevalence is significantly different, the algorithm should be recalculated. Fourth, to create the algorithm, we chose a cut-off point (10%). Still, the algorithm may be easily recreated with other cut-off points if these are considered more appropriate. Finally, we did not test the algorithm in general primary care. Therefore, readers should consider it experimental unless it shows effectiveness when properly tested in general primary care. It could be the case, for instance, that nurses working in a cardiac setting are more used to palpate and this may thus have influenced the accuracy of pulse palpation. That said, it may be worth noting that we did not find the sensitivity or specificity of the palpations to depend on the nurse's experience.

## Conclusions

Taken together, we propose optimal palpation of the pulse in individuals selected according to a straightforward algorithm to effectively detect AF in primary care. While the algorithm must still be tested in primary care settings, we believe that with these inexpensive recommendations, thousands of individuals may benefit from oral anticoagulation to prevent strokes, thus increasing their well-being and reducing the costs for the health system.

## Acknowledgments

We are indebted to the ten nurses who participated in the study in a totally altruistic way because they believed that we would improve the health of the population with the results. This paper is dedicated to the memory of Dr. David García-Dorado.

## Author Contributions

**Conceptualization:** Jordi Bañeras, Joaquim Radua.

**Data curation:** Ivana Pariggiano, Eduard Ródenas-Alesina, Gerard Oristrell, Roxana Escalona, Berta Miranda, Pau Rello, Toni Soriano, Blanca Gordon, Yassin Belahnech.

**Formal analysis:** Jordi Bañeras, Joaquim Radua.

**Investigation:** Jordi Bañeras, Joaquim Radua.

**Methodology:** Jordi Bañeras, David García-Dorado, Joaquim Radua.

**Project administration:** Jordi Bañeras, Joaquim Radua.

**Resources:** Jordi Bañeras, David García-Dorado, Joaquim Radua.

**Software:** Jordi Bañeras.

**Supervision:** Jordi Bañeras, Paolo Calabrò, David García-Dorado, Ignacio Ferreira-González, Joaquim Radua.

**Validation:** Jordi Bañeras, Joaquim Radua.

**Visualization:** Jordi Bañeras, Joaquim Radua.

**Writing – original draft:** Jordi Bañeras, Joaquim Radua.

**Writing – review & editing:** Ivana Pariggiano, Eduard Ródenas-Alesina, Gerard Oristrell, Roxana Escalona, Berta Miranda, Pau Rello, Toni Soriano, Blanca Gordon, Yassin Belahnech, Paolo Calabrò, David García-Dorado, Ignacio Ferreira-González.

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
