## [Decision Letter · Decision Letter 0]

16 Nov 2021

PONE-D-21-32308Optimal opportunistic screening of atrial fibrillation in primary care: who and howPLOS ONE

Dear Dr. Bañeras,

Thank you for submitting your manuscript to PLOS ONE. After careful consideration, we feel that it has merit but does not fully meet PLOS ONE’s publication criteria as it currently stands. Therefore, we invite you to submit a revised version of the manuscript that addresses the points raised during the review process.

Please read carefully the reviewers' comments and improve your manuscript as for their suggestions.

We look forward to receiving your revised manuscript.

Kind regards,

Giulio Francesco Romiti

Academic Editor

PLOS ONE

Journal Requirements:

Reviewers' comments:

Reviewer's Responses to Questions

5. Review Comments to the Author

Reviewer #1: This is a timely study.

As the authors note, a limitation is that patients with atrial flutter or a PPM were excluded.

Can you please clarify what the term "doubtful" means. Does this mean it is doubtful that any pulse can be felt? Or does this mean it is doubtful that the patient has atrial fibrillation. It is important to the readers to know how you defined terms.

I also think the discussion can be expanded to make it clear that the reason screening is not in the US prevention guidelines is because no study to date has shown that initiation of therapy for screen detected AD improves outcomes as compared with no screening at all.

Reviewer #2: I understood the purpose of the research to study an optimal pulse palpation method and create an algorithm to select patients for whom pulse palpation is useful, and authors created the algorithm from the research results.　However, there is room for improvement in the overall structure of the paper, the use of terms, and the content of the abstract.

1) Doesn't the meaning of 'accuracy' mean 'diagnostic accuracy'? Does the meaning of 'undetectable' mean 'unable to palpate pulse'? Especially for the meaning of 'undetectable' , I cannot recognize clearly.

Also, four possible definitions of positive palpation are difficult to understand in page8. What was doubtful? Was the palpation of pulses weak, or difficult to decide regular or irregulat?　The definition of these terms should be explaind clearly.

2) I couldn't find in the text 'sensitivity = 79%' on the second line of Abstract Results, is that correct?

3) In the Abstract, it is better to describe the contents of the algorithm.

4) The 4th and subsequent lines on page 5 of the Introduction are suitable for the Discussion.

5) 'A first aim',' A second aim', and'A third aim' on pages 8-9 are the contents of the Introduction, and it is better to describe only the methods for these aims in 'Materials and Methods'.

6) The rationale for 3% of the'Creation of the algorithm' on page 9 is unclear.

7) Doesn't 'AF' in the last line of page 11 mean 'unknown AF'?

8) Isn't 'should caution' on line 15 on page 13 'should be careful for'?

9) Although the results of conventional research are introduced in 'Discussion', the interpretation of the results of Authors' research is not fully described in 'Discussion'. Introduction of previous research should be restricted to the research directly related to the purpose and results of Authors' research. The focus of 'Discussion' is ambiguous and it is difficult to recognize 　'what was considered' in Authors' Discussion. For example, the'First' limitation described from line 7 on page 14 is considered to be a statement that asserts the usefulness of the results of Authors' study rather than 'limitation'.

Reviewer #3: Thank you for the opportunity to read and review this manuscript titled “Optimal opportunistic screening of atrial fibrillation in primary care: who and how”. The manuscript is of interest particularly to readers engaged in AF screening, as well as those interested in public health initiatives and primary care. The manuscript evaluated different pulse palpation parameters for optimally screening for AF in people age 40 years or older in order to design a simple algorithm that could be transferable to primary care to help identify opportunistically patients that would benefit from having regular pulse palpation. The research was performed in 4 cardiology outpatient clinics using trained nurses who palpated 7844 pulses and then followed these with a 12-lead ECG. Basic clinical data was also assessed. The authors concluded that palpation of the radial artery was the most accurate parameter to use, however specificity decreased with age, whilst heart failure was the only clinical factor that required an adapted approach.

This is a nicely written manuscript of what must have been a huge undertaking and the information presented will hopefully improve detection of AF in primary care if adopted. I have a few comments and suggestions, which I have listed below.

In this manuscript the authors have not tested the designed algorithm in primary care, which is slightly disappointing, it would have been nice to confirm if the algorithm had been transferable and strengthened the work considerably. It would also have been a fantastic opportunity to test the accuracy of the more recently developed single-lead ECG devices to then compare with the pulse and 12-lead. Particularly as a number of studies have reported pulse palpation having a much lower specificity than the newer technology.

I find the title of the study slightly misleading. My initial thoughts when reading this was that this study had been performed in a primary care setting with a number of healthcare professionals. I would suggest altering the title to reflect the study that was performed.

In relation to this the fact that the study hadn’t been conducted in primary care for the obvious reasons that the authors state I would suggest being more circumspect as to whether this would be transferable to primary care. There have been a number of studies assessing the accuracy of pulse palpation with various HCPs, some in primary care, very few have been referred to in this manuscript.

Nurses working in a cardiac setting would be used to taking pulses and ECG’s this may thus have influenced the accuracy data of pulse palpation and may not truly reflect how this would work in primary care, where pulse palpation by most HCPs is less routine. I think this would be a limitation of the study, which should be noted as such.

I think it would also be helpful to know more details about the nurses involved in the study, the number of nurses involved and the median of their years of experience.

From the data as well as expressing the specificity and sensitivity, I would suggest defining the number of false positives and listing the causes of these (e.g. ectopic beats) and how this compares with other studies. This would make the data more accessible to the lay reader.

I think it would also have been nice to show the number of new opportunistic AF cases that were detected in this study, as this would have shown relevance to opportunistic pulse testing in all healthcare settings.

Although not the basis of the study, I think a table stating some of the baseline demographics of the participants would have been useful to the field. I accept that this may be a huge undertaking however, to organise.

Finally, I think it would have been nice to see some cost-effectiveness calculations and comparisons with other designed screening methods

Reviewer #4: This is a study exploring an interesting concept aimed at assisting clinicians in prioritising resources to detect AF amongst patients that are the most likely to present with the condition. The consideration of multiple parameters as part of the algorithm creation is certainly a major strength of this study. On the other hand, the selection of pulse palpation as an index test to detect AF in the age of digital technology is somewhat questionable. This is demonstrated by authors themselves in a form of limited diagnostic sensitivity and specificity of pulse palpation ascertained by the study compared to modern ECG-based diagnostic modalities discussed elsewhere. The recommendations pertaining to AF screening in individuals aged < 65 years of age that are drawn from the manuscript are questionable at best considering the pre-existing evidence and guideline recommendations. It is also difficult to generalise the findings from the cardiology clinic to general practice or primary care, and I would urge authors to change their stance in this attempt, focusing on patients attending the cardiology clinic instead.

1. “Optimal opportunistic screening of atrial fibrillation in primary care: who and how” Considering the fact that the paper is solely focused on pulse palpation in a cardiology clinic setting (rather than primary care), I suggest amending the title to “Optimal opportunistic screening of atrial fibrillation using pulse palpation in a cardiology clinic: who and how” or similar to that effect. I suggest removing “primary care” from this manuscript since it relates to screening in a secondary care or specialist outpatient clinic.

2. Abstract. A well-presented summary of the study.

“Methods: In 4 Cardiology outpatient clinics, 7844 pulses were palpated according to a randomized list of arterial territories and durations of measure, and immediately followed by a 12-lead ECG, which we used as the ground truth.” I suggest changing “the ground truth” to “reference standard” as per STARD guidance/checklist.

3. Introduction:

• “Atrial fibrillation (AF) is the most common arrhythmia,1 and its prevalence rises with age, from about 2% in the whole population to about in 10%-17% in individuals aged 80 years or older.2” Please kindly remove “in” after “about” (typo). Please also review reference No. 1 which does not relate to AF. I suggest changing to one of the following:

Lip G.Y.H., Kakar P. & Watson T. (2007) Atrial fibrillation--the growing epidemic. Heart (British Cardiac Society) 93(5), 542-543.

Or Chugh S.S., Havmoeller R., Narayanan K., Singh D., Rienstra M., Benjamin E.J., et al. (2014a) Worldwide Epidemiology of Atrial Fibrillation: A Global Burden of Disease 2010 Study. Circulation 129(8), 837-847.

• “However, some voices have raised criticism about current screenings for AF.” Suggest changing to “However, some voices have raised criticism about the current AF screening initiatives.”

• “We hope that physicians might successfully apply this algorithm to select the individuals to who optimally palpate the pulse, efficiently improving the detection of AF and thus the prevention of stroke and its costs.” Suggest changing to “We hope that physicians might successfully apply this algorithm to select individuals in whom to optimally palpate the pulse, efficiently improving the detection of AF and thus the prevention of stroke and its costs.”

• Whilst the two paragraphs on page 5 are very informative and provide a rationale for the study, they partially overlap with the Methods section and could perhaps be made shorter. Instead, authors may consider including a statement or a sentence relating to the aim/objectives of the study.

4. Materials and Methods:

• It is possible that the definition of “primary care” varies between the healthcare systems, however from authors’ description “primary care” appears to be more resemblant of specialist or secondary/tertiary care than the traditional general practice or family medicine.

• The operator of the test was blinded to the ECG result, which helps ensure the validity of diagnostic accuracy data.

• It would be interesting to know why authors selected right radial/carotid pulse palpation and did not include other options, such as ulnar pulse which may sometimes be used in practice. Similarly, no rationale is provided for the selected duration of pulse palpation, i.e. 10 seconds vs. the commonly used 15, 30 or 60 seconds.

• “Immediately after pulse palpation, the nurse conducted a 12-lead ECG, considered the gold standard for detecting AF after reading by a cardiologist,14 and which we used as ground truth.” Suggest changing to ““Immediately after pulse palpation, the nurse conducted a 12-lead ECG, considered to be the gold standard for detecting AF after reading by a cardiologist,14 and which we used as a reference standard.”

• The diagnostic categories of pulse palpation appear overly complex. It may have been easier to only ask nurses to differentiate between: regular, irregular, doubtful or undetectable. It may have also made data analysis somewhat more straightforward.

• An excellent selection of independent variables for the mixed-effects model, covering most important bases.

5. Results

• It is very valuable to see the associations between diagnostic accuracy measures and various parameters (such as patient’s age), however it would also be helpful to include a conventional Table 1, that may contain the demographic details of all study participants (including their average age, comorbidities, etc.). The fact that 18.4% of patients had AF suggests that this study sample was perhaps quite different from what one may expect in the general population, including primary care.

• “The duration of palpation was 10 seconds in 49.9% palpations, and the necessary time in 50.1%.” It would be useful to know what the average “necessary time” was because that may have some effect on the utilisation of human resources, and possibly diagnostic accuracy (although the results presented here show otherwise).

• “Specificity was statistically significantly lower in older individuals (89% in <73 years and 83% in ≥73 years, p<0.001).” Why was the age threshold for older individuals set at 73 or 70 years and not the conventional 65 years or even 75 years? This would be relevant to appropriate international guidelines for opportunistic AF screening. Apart from PPV/FOR, did participant’s age affect the sensitivity or specificity of the test?

• Table 2. I can see the rationale for conducting this analysis and for including this table. It does however take the reader some time to review all data included, and it probably warrants a brief summary paragraph to explain the key findings at a glance, especially since the purpose of this study was to develop an algorithm. One may also argue that the analysis which include participants under the age of 65 years is somewhat less beneficial considering the fact that few of them would actually have AF and that most of those with detectable cases would not require interventions, such as anticoagulation. Indeed, few studies have ever evaluated the diagnostic accuracy and/or cost-effectiveness of screening individuals < 65 years of age for AF.

• “Despite pulse palpation has been widely recommended to screen for AF, there is little evidence about which are the optimal palpation parameters or about its positive predictive values and false omission rates in general practice settings.” Suggest changing to “Despite pulse palpation being widely recommended to screen for AF, there is little evidence about the optimal palpation parameters or about its positive predictive values and false omission rates in general practice settings.”

6. Discussion and Conclusions

• “For example, single-time-point ECG recording in a general population ≥65 years of age detected AF in 1.4%.” I suggest making it clear these data are from a systematic review rather than primary literature.

• “A straightforward strategy to improve the efficiency of the detection of AF consists in first selecting those individuals with higher risk of AF, and the cheapest and quickest method to evaluate this risk is palpation of the pulse.” I suggest rewording to “A straightforward strategy to improve the efficiency of AF detection consists of first selecting those individuals with a higher risk of AF, and the cheapest and quickest method to evaluate this risk is palpation of the pulse.”

• “Indeed, systematic pulse assessment during routine clinic visits followed by 12-lead ECG in those with an irregular pulse has been already reported to substantially increase the detection of AF.” That is correct, however this and many other studies have shown that opportunistic screening is likely to be even more effective and cost-effective than systematic approaches. This raises a question of whether the authors should have focused on creating an algorithm for pulse palpation in population groups for whom AF detection is already recommended, e.g. those aged 65 and over.

• “In any case, our estimates may be not directly comparable to those of previous studies because most of latter did not clearly reported how they included or coded doubtful and undetectable palpations.10” Please change “reported” to “report”.

• “Morgan et al found also found high specificity (98%) but low sensitivity (54%) when considering an abnormal pulse as a ‘continuous irregularity’.21” Remove one of the “found”. Also, authors may wish to state the fact their results were perhaps closer to those of “frequent or continuous irregularity” defined by Morgan & Mant (sensitivity 72%, specificity 94%), which makes sense considering the criteria used by the present study for the positive test result.

• Reference 22. I could not relate this reference to data regarding the sensitivity and specificity of AF detection presented by authors. They may wish to review the reference selected accordingly.

• “The importance of age in AF may has led many screening studies to use a threshold of ≥ 65 years.21,22” Please change “has” to “have”.

• “Younger people can also receive anticoagulation if they meet the CHADS-VASc score criteria, because 1-2 points of the criteria are age>65-75 years, but the other 7 points are not age-related.12” This statement is correct, however the low prevalence of comorbidities included in the CHA2DS2-VASc score (e.g. heart failure) amongst younger patients, means that the majority of < 65-year olds may not benefit from AF screening. Please change “CHADS-VASc” to “CHA2DS2-VASc”.

• “In the recent years, there have appeared new AF detection methods based on the use of novel technologies, such as smartphones with ECG electrodes, smart watches, or blood pressure machines.” It is crucial not to combine single- or multiple-lead ECG devices and photoplethysmography-based devices/applications under the same umbrella: not just because the former may offer additional diagnostic potential for AF (30-second ECG is diagnostic of AF), but also because of differences in diagnostic accuracy (particularly specificity). I would urge caution when combining these technologies together considering the growing amount of evidence (including studies of cost-effectiveness) that supports the use of ECG-based devices in AF screening.

• “Conversely, our proposed screening would consist in a simple and inexpensive pulse palpation, which does not raise anxiety and is acceptable to most individuals.10” Suggest changing to “consist of”.

• “However, the sensitivity and specificity of a screening tool do not depend on the prevalence of the medical condition, and we checked that they neither depended on a range of heart conditions, so that we can safely translate the results to the general population.” It is somewhat unusual that the presence of heart failure influenced the PPV/FOR but not the sensitivity or specificity. Am I correct in confirming this and what could be the underlying reason(s)?

• “However, the sensitivity and specificity of a screening tool do not depend on the prevalence of the medical condition, and we checked that they neither depended on a range of heart conditions, so that we can safely translate the results to the general population.” As previously mentioned, despite the adjustments, the generalisation of these findings requires an adequate caution, considering they were drawn from cardiology patients and not patients in general practice.

• Overall, the conclusion mostly draws on the findings of the study. On the other hand, at present I cannot support the use of term “primary care” instead of “outpatient” or “cardiology” clinic, taking into account the setting and nature of patients involved. Similarly, in the absence of substantial evidence, I would question the benefits of screening some patients aged < 65 years of age which is encouraged by authors of the study.

---

## [Author Response · Author response to Decision Letter 0]

3 Feb 2022

We are very grateful to each of the reviewers. The comments have undoubtedly improved the scientific quality of the manuscript.

RESPONSES TO REVIEWER #1:

This is a timely study. As the authors note, a limitation is that patients with atrial flutter or a PPM were excluded.

Response: Thank you very much for the review. We have followed all suggestions, and the text has definitely improved thanks to your comments.

1. Can you please clarify what the term "doubtful" means. Does this mean it is doubtful that any pulse can be felt? Or does this mean it is doubtful that the patient has atrial fibrillation. It is important to the readers to know how you defined terms.

Response: Thank you for noting this imprecision in the text. We have replaced "doubtful" with "uncertain about pulse regularity" throughout the abstract, text, and tables. To further clarify, we have also replaced "undetectable" with "unable to palpate".

2. I also think the Discussion can be expanded to make it clear that the reason screening is not in the US prevention guidelines is because no study to date has shown that initiation of therapy for screen detected AD improves outcomes as compared with no screening at all.

Response: Thank you very much for your observation, which clearly increases the whole perspective regarding treatment. In this sense, as suggested, we have added a new paragraph to the discussion, as well as 2 bibliographic citations.

“Finally, regarding to treatment, we do not know whether silent AF episodes require anticoagulation therapy, since no trials have assessed the benefits and harms of anticoagulation treatment among screen-detected populations. In this context, the United States Preventive Services Task Force concludes that the current evidence is insufficient to assess the balance of benefits and harms of screening for atrial fibrillation (32). However, In the ASSERT trial subclinical AF was associated with an increased risk of clinical AF (HR 5.56; 95% CI 3.78–8.17; P < 0.001) and of ischaemic stroke or systemic embolism (HR 2.49; 95% CI 1.28–4.85; P = 0.007). In the next years, several trials should be able to tell us if anticoagulation therapy should be used for short-lasting episodes diagnosed on cardiac devices (33)”. (page 15)

32. Kahwati L, Asher GN, Kadro Z, et al. Atrial Fibrillation Screening: A Review of the Evidence for the US Preventive Services Task Force https://www.uspreventiveservicestaskforce.org/uspstf/document/draft-evidence-review/screening-atrial-fibrillation. Retrieved December 28, 2021.

33. Camm AJ, Simantirakis E, Goette A, Lip GY, Vardas P, Calvert M, Chlouverakis G, Diener HC, Kirchhof P. Atrial high-rate episodes and stroke prevention. Europace. 2017 Feb 1;19(2):169-179. doi: 10.1093/europace/euw279. 

RESPONSES TO REVIEWER #2:

I understood the purpose of the research to study an optimal pulse palpation method and create an algorithm to select patients for whom pulse palpation is useful, and authors created the algorithm from the research results.　However, there is room for improvement in the overall structure of the paper, the use of terms, and the content of the abstract.

Response: Thank you very much for the review. We have followed all suggestions, and we think that the text has improved.

1. Doesn't the meaning of 'accuracy' mean 'diagnostic accuracy'? Does the meaning of 'undetectable' mean 'unable to palpate pulse'? Especially for the meaning of 'undetectable', I cannot recognize clearly. Also, four possible definitions of positive palpation are difficult to understand in page 8. What was doubtful? Was the palpation of pulses weak, or difficult to decide regular or irregular?　The definition of these terms should be explained clearly.

Response: We thank the reviewer for his suggestions for improving the terms, which we have applied to make the definitions clearer. Specifically, we have replaced "accuracy" with "diagnostic accuracy", we have replaced "undetectable" by "unable to palpate", and we have substituted "doubtful" with "uncertain about pulse regularity" throughout the abstract, text, and tables. Finally, to make the four possible definitions of positive palpation more understandable, we have rephrased the text as follows: " a) only when the nurse coded the pulse as irregular; b) when the nurse coded the pulse as irregular or was uncertain about its regularity; c) when the nurse coded the pulse as irregular or was unable to palpate it; d) when the nurse coded the pulse as irregular, was uncertain about its regularity, or was unable to palpate it." (page 8).

2. I couldn't find in the text' sensitivity = 79%' on the second line of Abstract Results, is that correct?

Response: Thank you sincerely for spotting this inconsistency. In the previous version, we had written in the text and table the estimations from the model that compared the three arterial territories. In contrast, we had written in the abstract the estimation from the model using only radial palpations. They differed slightly (78% vs. 79%). Therefore, to be consistent, we now only provide the estimations from the models using only carotid palpations, only radial palpations, or both (as we had previously done for the Abstract).

 “ Specifically, radial palpations showed 79% sensitivity, 86% specificity, and 83% balanced diagnostic accuracy” (Page 10).

3. In the Abstract, it is better to describe the contents of the algorithm.

Response: Thank you for this suggestion. We now describe the contents of the algorithm: " a) do nothing in <60 years old individuals without heart failure; b) do ECG in ≥70 years old individuals with heart failure; c) do radial pulse palpation in the remaining individuals and do ECG if the pulse is irregular or you are uncertain about its regularity or unable to palpate it.".

4. The 4th and subsequent lines on page 5 of the Introduction are suitable for the Discussion.

Response: We have moved this text to the beginning of the Discussion (page 11-12):

“To shed light on this field, we report the results of a large study to assess the diagnostic accuracy (sensitivity and specificity) of pulse palpation as a screening tool to detect AF depending on palpation parameters and clinical factors. To include many individuals with AF, we conducted the study in cardiology outpatients, in whom the prevalence of AF is higher than in the general population. As we detailed earlier, we checked that sensitivity and specificity did not depend on our sample's range of heart conditions. Thus, our results should be generalizable to the general population. However, we must note that if we had conducted this study in a sample from the general population, the number of individuals with AF would have been substantially smaller. Thus, the estimates of sensitivity would be imprecise. That said, we encourage a future study in general primary care patients to ensure the transferability of the results of this study.

Wee estimated the PPV and the FOR, i.e., the probabilities of having AF if pulse palpation is positive or negative in individuals with different clinical characteristics. PPV and FOR allowed us to create a simple clinical algorithm based on very few clinical factors to help know which individuals may benefit from a palpation screening in primary care. For example, there are few reasons to conduct the screening in individuals who will be very unlikely to have AF even if pulse palpation is positive or in individuals who will still have a significant probability of having AF even if palpation is negative. Conversely, there are clear reasons to conduct the screening in individuals with a significant probability of having AF if palpation is positive and a negligible probability if palpation is negative”. 

5. 'A first aim',' A second aim', and 'A third aim' on pages 8-9 are the contents of the Introduction, and it is better to describe only the methods for these aims in 'Materials and Methods'.

Response: We have moved the aims to the end of the Introduction.

6. The rationale for 3% of the 'Creation of the algorithm' on page 9 is unclear.

Response: We agree and now use 10%, a rounder number: " with PPV<10% because PPV<10% means that they would have a low probability of AF (<10%) even if palpation were positive. Similarly, we considered recommending ECG without the need for prior palpation in individuals with FOR>10% because FOR>10% means that they would have a high probability of AF (>10%) even if palpation were negative. We deemed it appropriate for the remaining individuals to conduct pulse palpation, followed by an ECG when positive. " (page 9).

In addition, we make clearer that “It is important to remark that this cut-off point (10%) is only a suggestion, and other cut-off points may be more appropriate in a specific health system. Fortunately, the algorithm may be easily recreated with other cut-off points.” (page 10)

7. Doesn't 'AF' in the last line of page 11 mean 'unknown AF'?

Response: Thank you for noting this imprecision. We have added "unknown" to clarify it (page 12).

8. Isn't 'should caution' on line 15 on page 13 'should be careful for'?

Response: Thank you for the suggestion. We have rephrased it as follows: " we should be careful about " (page 14)

9. Although the results of conventional research are introduced in 'Discussion', the interpretation of the results of Authors' research is not fully described in 'Discussion'. Introduction of previous research should be restricted to the research directly related to the purpose and results of Authors' research. The focus of 'Discussion' is ambiguous and it is difficult to recognize 'what was considered' in Authors' Discussion. For example, the 'First' limitation described from line 7 on page 14 is considered to be a statement that asserts the usefulness of the results of Authors' study rather than 'limitation'.

Response: We agree with this comment, so we have re-ordered the text and have given more emphasis to the results of our study. The fact that previous studies have not taken into account that in reality the results of pulse palpation can be classified as "uncertain about pulse regularity" or "with" unable to palpate ", makes comparisons with previous studies difficult.

We have eliminated the first limitation, since in fact the reason for choosing the sample is justified at the beginning of the discussion.

Based on your suggestions and those of the other reviewers, we have made changes to the introduction, discussion, and limitations. We think that the discussion now follows a more logical sequence, ending with the final discussion on anticoagulant treatment.

RESPONSES TO REVIEWER #3:

Thank you for the opportunity to read and review this manuscript titled "Optimal opportunistic screening of atrial fibrillation in primary care: who and how". The manuscript is of interest particularly to readers engaged in AF screening, as well as those interested in public health initiatives and primary care. The manuscript evaluated different pulse palpation parameters for optimally screening for AF in people age 40 years or older in order to design a simple algorithm that could be transferable to primary care to help identify opportunistically patients that would benefit from having regular pulse palpation. The research was performed in 4 cardiology outpatient clinics using trained nurses who palpated 7844 pulses and then followed these with a 12-lead ECG. Basic clinical data was also assessed. The authors concluded that palpation of the radial artery was the most accurate parameter to use, however specificity decreased with age, whilst heart failure was the only clinical factor that required an adapted approach.

This is a nicely written manuscript of what must have been a huge undertaking and the information presented will hopefully improve detection of AF in primary care if adopted. I have a few comments and suggestions, which I have listed below.

Response: Thank you very much for the review. We have followed all suggestions, and we think that the text has improved.

1. In this manuscript the authors have not tested the designed algorithm in primary care, which is slightly disappointing, it would have been nice to confirm if the algorithm had been transferable and strengthened the work considerably. It would also have been a fantastic opportunity to test the accuracy of the more recently developed single-lead ECG devices to then compare with the pulse and 12-lead. Particularly as a number of studies have reported pulse palpation having a much lower specificity than the newer technology.

Response: We agree with the reviewer that testing the algorithm in primary care or the accuracy of the single-lead ECG devices would be great and have now included these limitations: " Second, we did not investigate the accuracy of more recently developed single-lead ECG devices. Future studies might investigate the effectiveness of these devices in detecting unknown AF in primary care. " (page 15) and " Finally, we did not test the algorithm in general primary care. Therefore, readers should consider it experimental unless it shows effectiveness when properly tested in general primary care." (page 16).

2. I find the title of the study slightly misleading. My initial thoughts when reading this was that this study had been performed in a primary care setting with a number of healthcare professionals. I would suggest altering the title to reflect the study that was performed.

Response: We agree that the title was confusing. We have replaced "in primary care" with "in cardiology outpatient clinics" to avoid confusion.

New title: “Optimal opportunistic screening of atrial fibrillation using pulse palpation in cardiology outpatient clinics: who and how” (page 1 and 2).

3. In relation to this the fact that the study hadn't been conducted in primary care for the obvious reasons that the authors state I would suggest being more circumspect as to whether this would be transferable to primary care.

Response: In addition to the new limitation detailed in answer to comment 1, we have now included several sentences in this regard: " That said, we encourage a future study in general primary care patients to ensure the transferability of the results of this study" (page 11) and " Taken together, we propose optimal palpation of the pulse in individuals selected according to a straightforward algorithm to effectively detect AF in primary care. While the algorithm must still be tested in primary care settings, " (page 16).

4. There have been a number of studies assessing the accuracy of pulse palpation with various HCPs, some in primary care, very few have been referred to in this manuscript.

Response: We have added to the classical studies of Fitzmaurice (reference 10), Morgan (reference 21) the study of Sudlow.

We have replace reference 22: “Kavanagh S, Knapp M. The impact on general practitioners of the changing balance of care for elderly people living in institutions. BMJ. 1998;317(7154):322-327. doi:10.1136/bmj.317.7154.322” for Sudlow M, Rodgers H, Kenny RA, Thomson R. Identification of patients with atrial fibrillation in general practice: a study of screening methods. BMJ. 1998;317(7154):327-328. doi:10.1136/bmj.317.7154.327

The reader has 3 references on systematic reviews (14-Screening strategies for atrial fibrillation: a systematic review and cost-effectiveness analysis 18- Screening to identify unknown atrial fibrillation: A systematic review. 20- Accuracy of methods for detecting an irregular pulse and suspected atrial fibrillation: A systematic review and meta-analysis) that contain the vast majority of studies on pulse plapation screening. It should be noted that screening with new technologies has displaced interest in classical screening by physical examination.

5. Nurses working in a cardiac setting would be used to taking pulses and ECG's this may thus have influenced the accuracy data of pulse palpation and may not truly reflect how this would work in primary care, where pulse palpation by most HCPs is less routine. I think this would be a limitation of the study, which should be noted as such.

Response: We have added this limitation: " Finally, we did not test the algorithm in general primary care. Therefore, readers should consider it experimental unless it shows effectiveness when properly tested in general primary care. It could be the case, for instance, that nurses working in a cardiac setting are more used to palpate and this may thus have influenced the accuracy of pulse palpation. That said, it may be worth noting that we did not find the sensitivity or specificity of the palpations to depend on the nurse's experience." (page 16).

6. I think it would also be helpful to know more details about the nurses involved in the study, the number of nurses involved and the median of their years of experience.

Response: We did not specifically ask for years of experience for confidentiality reasons, but all of them held meetings with the principal investigator. None had previously received workshops on how to palpate the pulse or applied it in their regular clinical practice. In relation to the years of experience, the vast majority had years of experience as nurses in outpatient clinics, but none were specific to cardiology, since according to the week's program they were with different specialties (ophthalmology, general surgery, dermatology, endocrinology, etc.)

So that the reader receives more information, and can better interpret the results, where we report " the duration of palpation and the nurse's experience did not show statistically significant effects on specificity, and no palpation parameters showed statistically significant effects on sensitivity.", we have added " The least frequent pulse palpations per nurse were 36 and 210, and the most frequent palpations per nurse were 1696 and 2560" to the text.(page 10)

7. From the data as well as expressing the specificity and sensitivity, I would suggest defining the number of false positives and listing the causes of these (e.g. ectopic beats) and how this compares with other studies. This would make the data more accessible to the lay reader.

Response: The design did not take into account the review of false positives, since from the moment the pulse was palpated until the ECG recording was obtained the heart rate could change or the extrasystole disappear as an example of modifying factors.

In this sense, false-positive results are mainly because of premature ectopic beats (Proesmans T, Mortelmans C, Van Haelst R, Verbrugge F, Vandervoort P, Vaes B. Mobile Phone-Based Use of the Photoplethysmography Technique to Detect Atrial Fibrillation in Primary Care: Diagnostic Accuracy Study of the FibriCheck App. JMIR Mhealth Uhealth. 2019;7(3):e12284. Published 2019 Mar 27. doi:10.2196/12284)

So we do not know what component of the false positive depends on the nurse and what component of the rhythm that the patient presents. In any case, we do know that all the nurses did not show differences in specificity, and therefore, in the rate of false positives, in their measurements.

8. I think it would also have been nice to show the number of new opportunistic AF cases that were detected in this study, as this would have shown relevance to opportunistic pulse testing in all healthcare settings.

Response: Through screening, 131 new cases of atrial fibrillation were detected (3.2% of patients), but we do not report this data because it can lead to misinterpretation, since it must be taken into account that the screening was done in outpatient cardiology clinics, where the predisposition to develop atrial fibrillation is much higher than in other external medical consultations. In any case our results are in line with the SAFE study, where more diagnoses of atrial fibrillation were detected by screening than usual care (1.63% v 1.04%, difference=0.59%, 95% confidence interval 0.20 to 0.98).

On the other hand, it must also be taken into account that there is a factor that cannot be controlled, and that is a paroxysmal character of atrial fibrillation. In our study population, there were 9.5% of patients with a history of atrial fibrillation, but on the day of screening, only 8.8% of them were in atrial fibrillation.

9. Although not the basis of the study, I think a table stating some of the baseline demographics of the participants would have been useful to the field. I accept that this may be a huge undertaking however, to organise.

Response: Thank you for this suggestion. We now provide this table (new Table 1).

 Individuals Palpations

 All Sinusoidal rhythm Atrial fibrillation Both(a) All Sinusoidal rhythm Atrial fibrillation

Sample size 5,508 4,490 922 96 7,430 6,060 1,370

Age (mean±SD) 70±13 69±14 77±8 77±9 71±13 69±13 77±8

Females (%) 43% 42% 48% 43% 42% 41% 47%

Body mass index (mean±SD) 29±5 29±5 30±5 29±5 29±5 29±5 30±5

Heart rate (mean±SD) 69±13 68±11 75±16 71±14 69±13 68±12 74±18

Ejection fraction (mean±SD) 57±10 57±10 56±10 57±10 57±10 57±10 56±10

High blood pressure (%) 74% 72% 83% 85% 74% 72% 84%

Dyslipidemia (%) 57% 58% 51% 64% 58% 59% 53%

Diabetes mellitus (%) 26% 25% 26% 36% 27% 26% 28%

Peripheral artery disease (%) 7% 7% 5% 11% 8% 8% 8%

Hypertensive heart disease (%) 34% 32% 44% 33% 34% 32% 44%

Ischemic cardiomyopathy (%) 40% 43% 25% 39% 43% 46% 29%

Right bundle branch block (%) 8% 8% 6% 8% 8% 8% 6%

Left bundle branch block (%) 5% 5% 5% 5% 5% 5% 6%

10. Finally, I think it would have been nice to see some cost-effectiveness calculations and comparisons with other designed screening methods

Response: Thank you very much for your suggestion, which would certainly increase the impact of the results. We will work on that for a potential next publication.We have made a very first approximation to be able to carry out an economic analysis, but due to the amount of information we think it is more appropriate to write a separate article, to present potential results and carry out a discussion focused on this topic, now that there are many technological alternatives to compare.

RESPONSES TO REVIEWER #4:

This is a study exploring an interesting concept aimed at assisting clinicians in prioritising resources to detect AF amongst patients that are the most likely to present with the condition. The consideration of multiple parameters as part of the algorithm creation is certainly a major strength of this study. On the other hand, the selection of pulse palpation as an index test to detect AF in the age of digital technology is somewhat questionable. This is demonstrated by authors themselves in a form of limited diagnostic sensitivity and specificity of pulse palpation ascertained by the study compared to modern ECG-based diagnostic modalities discussed elsewhere. The recommendations pertaining to AF screening in individuals aged < 65 years of age that are drawn from the manuscript are questionable at best considering the pre-existing evidence and guideline recommendations. It is also difficult to generalise the findings from the cardiology clinic to general practice or primary care, and I would urge authors to change their stance in this attempt, focusing on patients attending the cardiology clinic instead.

Response: Thank you very much for the review. We have followed all suggestions and we think that the text has improved. Regarding the selection of pulse palpation in the age of digital technology, we agree on the potential of the new technology, but we think that there are still aspects to be solved, because for example for the same device (AliveCor) in the study published by CHan et al, a sensitivity of 71.4% is reported, and yet in the study of Lowres et al. A sensitivity of 98.5% is reported, this discrepancy in sensitivities being very remarkable. On the other hand, cost-benefit and cost-effectiveness studies are necessary, especially because access to new technologies is limited. In addition, as side effects, the false positive rate of the devices can have a negative impact on the mental health of patients. Until all these issues are resolved, we think that our study has a reason to be.

Regarding the benefits of screening patients <65 years old, the paper now shows that palpation is not recommended for individuals <60 years unless they have heart failure. However, for 60-64 individuals, we would recommend palpation even if they do not have heart failure, given that the PPV is high (around 25%). On the other hand, the age limit of 65 years is a threshold that has passed from classical studies to those studying new technologies. It should be borne in mind that there are other risk factors that play an important role in the development of a stroke. We think that further studies should think about including earlier ages, where there are a potential number of patients who would also benefit from screening if the patient is affected by other recognized risk factors. Although the CHA2DS2-VASc is used to decide whether to anticoagulate, screening should also be contextualized.

Finally, regarding the generalization to general primary care, we have changed the title and parts of the text to make it more transparent that our study was conducted in cardiology outpatient clinics. We expand these changes below in the responses to the specific comments.

1. "Optimal opportunistic screening of atrial fibrillation in primary care: who and how" Considering the fact that the paper is solely focused on pulse palpation in a cardiology clinic setting (rather than primary care), I suggest amending the title to "Optimal opportunistic screening of atrial fibrillation using pulse palpation in a cardiology clinic: who and how" or similar to that effect. I suggest removing "primary care" from this manuscript since it relates to screening in a secondary care or specialist outpatient clinic.

Response: We agree that the title was confusing. We have replaced "in primary care" with "using pulse palpation in cardiology outpatient clinics" to avoid confusion. 

New title: “Optimal opportunistic screening of atrial fibrillation using pulse palpation in cardiology outpatient clinics: who and how” (page 1 and 2).

We have also added some sentences about the transferability of the results to the general primary care: " That said, we encourage a future study in general primary care patients to ensure the transferability of the results of this study" (page 11) and " Finally, we did not test the algorithm in general primary care. Therefore, readers should consider it experimental unless it shows effectiveness when properly tested in general primary care. It could be the case, for instance, that nurses working in a cardiac setting are more used to palpate and this may thus have influenced the accuracy of pulse palpation. That said, it may be worth noting that we did not find the sensitivity or specificity of the palpations to depend on the nurse's experience." (page 16).

2. Abstract. A well-presented summary of the study.

"Methods: In 4 Cardiology outpatient clinics, 7844 pulses were palpated according to a randomized list of arterial territories and durations of measure, and immediately followed by a 12-lead ECG, which we used as the ground truth." I suggest changing "the ground truth" to "reference standard" as per STARD guidance/checklist.

Response: Thank you for the suggestion. We have replaced "ground truth" with "reference standard" as suggested in the Abstract, page 7 and Table 2 footnote.

3. Introduction:

3.1. "Atrial fibrillation (AF) is the most common arrhythmia,1 and its prevalence rises with age, from about 2% in the whole population to about in 10%-17% in individuals aged 80 years or older.2" Please kindly remove "in" after "about" (typo). Please also review reference No. 1 which does not relate to AF. I suggest changing to one of the following:

Lip G.Y.H., Kakar P. & Watson T. (2007) Atrial fibrillation--the growing epidemic. Heart (British Cardiac Society) 93(5), 542-543.

Or Chugh S.S., Havmoeller R., Narayanan K., Singh D., Rienstra M., Benjamin E.J., et al. (2014a) Worldwide Epidemiology of Atrial Fibrillation: A Global Burden of Disease 2010 Study. Circulation 129(8), 837-847.

Response: Thank you for detecting the typo, which we have corrected, and for the references, which we have inserted in the place of the previous one (page 20) “Chugh S.S., Havmoeller R., Narayanan K., et al. (2014a) Worldwide Epidemiology of Atrial Fibrillation: A Global Burden of Disease 2010 Study. Circulation 129(8), 837-847.”

3.2. "However, some voices have raised criticism about current screenings for AF." Suggest changing to "However, some voices have raised criticism about the current AF screening initiatives."

Response: Thank you for the suggestion, which we have followed (page 5).

3.3. "We hope that physicians might successfully apply this algorithm to select the individuals to who optimally palpate the pulse, efficiently improving the detection of AF and thus the prevention of stroke and its costs." Suggest changing to "We hope that physicians might successfully apply this algorithm to select individuals in whom to optimally palpate the pulse, efficiently improving the detection of AF and thus the prevention of stroke and its costs."

Response: Thank you for the suggestion, which we have followed (at the end of page 5).

3.4. Whilst the two paragraphs on page 5 are very informative and provide a rationale for the study, they partially overlap with the Methods section and could perhaps be made shorter. Instead, authors may consider including a statement or a sentence relating to the aim/objectives of the study.

Response: We fully agree with the reviewer in improving the structure of this part. We have moved these paragraphs mostly to the beginning of the Discussion. Instead, we added a section with the aims at the end of the Introduction: " In this context, we investigated for the first time which pulse palpation parameters are more accurate to detect AF. We also estimated the predictive values to create an easy algorithm so that the primary care physician readily knows which individuals may benefit from this screening and which may not. Our first aim was to establish which palpation parameters would optimize the sensitivity and specificity of palpation. The second aim was to check whether the sensitivity or specificity of palpation might depend on a range of clinical factors. This checking was necessary for two reasons. On the one hand, we had to ensure that sensitivity and specificity did not depend on heart conditions. Thus, we could safely translate our results to the general population. On the other hand, we had to know the sensitivity and specificity of each algorithm stratum that we could create later. Finally, the final aim was to estimate the positive predictive values (PPV) and false omission rates (FOR) separately for strata of age, sex, and the presence of very few readily available clinical factors." (page 5).

4. Materials and Methods:

4.1. It is possible that the definition of "primary care" varies between the healthcare systems, however from authors' description "primary care" appears to be more resemblant of specialist or secondary/tertiary care than the traditional general practice or family medicine.

Response: As we detailed in response to comment 1, to avoid confusion, we have replaced "primary care" with "cardiology outpatient clinics" and added some sentences about the transferability of the results to general primary care.

4.2. The operator of the test was blinded to the ECG result, which helps ensure the validity of diagnostic accuracy data.

Response: Thank you.

4.3. It would be interesting to know why authors selected right radial/carotid pulse palpation and did not include other options, such as ulnar pulse which may sometimes be used in practice. Similarly, no rationale is provided for the selected duration of pulse palpation, i.e. 10 seconds vs. the commonly used 15, 30 or 60 seconds.

Response: Thank you very much for your observation. Unfortunately, the screening studies for atrial fibrillation with pulse palpation do not report the palpation methodology, so when interpreting the results, we do not know what territory or how long they used. In this sense, in our design we defined 2 territories: radial and carotid, because they were the 2 territories that by consensus of the group were interpreted as viable in an outpatient patient due to accessibility issues (the femoral pulse was ruled out for this reason). We did not take into account the ulnar palpation, due to lack of practice in our environment.

In relation to time, in the absence of a description in previous studies, in a pilot test we saw that it was a relevant aspect, since there were professionals who looked at the clock (we thought that it could be a confusing factor and that an irregular pulse was interpreted as regular by synchrony) and others who did not look at the clock but who required varying time duration. For this reason we decided to randomize the 2 strategies (free time vs clock). By consensus of the 10 nurses, it was established that 10 seconds would be sufficient, so we did not explore other times.

To clarify, we added: “10 seconds, defined by nursing consensus” (page 7)

4.4. "Immediately after pulse palpation, the nurse conducted a 12-lead ECG, considered the gold standard for detecting AF after reading by a cardiologist,14 and which we used as ground truth." Suggest changing to “"Immediately after pulse palpation, the nurse conducted a 12-lead ECG, considered to be the gold standard for detecting AF after reading by a cardiologist(14), and which we used as a reference standard."

Response: Thank you for the suggestion, which we have followed (page 7).

4.5. The diagnostic categories of pulse palpation appear overly complex. It may have been easier to only ask nurses to differentiate between: regular, irregular, doubtful or undetectable. It may have also made data analysis somewhat more straightforward.

Response: We agree that the definitions were complex. We have fully rephrased this part to make them more understandable: " a) only when the nurse coded the pulse as irregular; b) when the nurse coded the pulse as irregular or was uncertain about its regularity; c) when the nurse coded the pulse as irregular or was unable to palpate it; d) when the nurse coded the pulse as irregular, was uncertain about its regularity, or was unable to palpate it." (page 8).

4.6. An excellent selection of independent variables for the mixed-effects model, covering most important bases.

Response: Thank you.

5. Results

5.1. It is very valuable to see the associations between diagnostic accuracy measures and various parameters (such as patient's age), however it would also be helpful to include a conventional Table 1, that may contain the demographic details of all study participants (including their average age, comorbidities, etc.). The fact that 18.4% of patients had AF suggests that this study sample was perhaps quite different from what one may expect in the general population, including primary care.

Response: Thank you for this suggestion. We now provide this table (new Table 1).

5.2. "The duration of palpation was 10 seconds in 49.9% palpations, and the necessary time in 50.1%." It would be useful to know what the average "necessary time" was because that may have some effect on the utilisation of human resources, and possibly diagnostic accuracy (although the results presented here show otherwise).

Response: Thank you for your appreciation and rigor in methodology, which has been lacking in previous pulse palpation studies. Unfortunately we do not have the time duration used in those measurements that were taken with the necessary time, since the purpose was to see if looking at the clock or not there was a certain effect of perception of synchronization and regularity.

To clarify this aspect, we have added to the text: “With the strategy of preset time vs necessary time, we wanted to incorporate the variable of whether looking at a clock (10 seconds) had an effect on the perception of regularity due to synchronization phenomena and decreased the effectiveness of detecting AF.” (page 7)

5.3. "Specificity was statistically significantly lower in older individuals (89% in <73 years and 83% in ≥73 years, p<0.001)." Why was the age threshold for older individuals set at 73 or 70 years and not the conventional 65 years or even 75 years? This would be relevant to appropriate international guidelines for opportunistic AF screening. Apart from PPV/FOR, did participant's age affect the sensitivity or specificity of the test?

Response: We had defined the two age groups according to its median, but we have now defined according to the conventional 65 years: " age (<65 vs. ≥65 year old)," (page 8). We found that participants' age affected the test's specificity: " Specificity was statistically significantly lower in older individuals (91% in <65 years and 84% in ≥65 years, p<0.001). " (page 11).

5.4. Table 2. I can see the rationale for conducting this analysis and for including this table. It does however take the reader some time to review all data included, and it probably warrants a brief summary paragraph to explain the key findings at a glance, especially since the purpose of this study was to develop an algorithm. One may also argue that the analysis which include participants under the age of 65 years is somewhat less beneficial considering the fact that few of them would actually have AF and that most of those with detectable cases would not require interventions, such as anticoagulation. Indeed, few studies have ever evaluated the diagnostic accuracy and/or cost-effectiveness of screening individuals < 65 years of age for AF.

Response: We have fully recreated the table (now Table 2) to make it more straightforward. Specifically, we now provide the PPV and FOR in the 5-year age ranges, along with an explanation. In addition, we have repeated the analysis using 5-year ranges (instead of 10-year ranges) and 10% cut-off (instead of 3% cut-off). It shows that palpation is not recommended for individuals <60 years unless they have heart failure. However, for 60-64 individuals, we would recommend palpation even if they do not have heart failure, given that the PPV is high (around 25%).

5.5. "Despite pulse palpation has been widely recommended to screen for AF, there is little evidence about which are the optimal palpation parameters or about its positive predictive values and false omission rates in general practice settings." Suggest changing to "Despite pulse palpation being widely recommended to screen for AF, there is little evidence about the optimal palpation parameters or about its positive predictive values and false omission rates in general practice settings."

Response: Thank you for the suggestion, which we have followed (page 11).

6. Discussion and Conclusions

6.1. "For example, single-time-point ECG recording in a general population ≥65 years of age detected AF in 1.4%." I suggest making it clear these data are from a systematic review rather than primary literature.

Response: We have rephrased the sentence to clarify: " a previous systematic review found that single-time-point ECG recording detected unknown AF in 1.4% ≥65-year- individuals " (page 12).

6.2. "A straightforward strategy to improve the efficiency of the detection of AF consists in first selecting those individuals with higher risk of AF, and the cheapest and quickest method to evaluate this risk is palpation of the pulse." I suggest rewording to "A straightforward strategy to improve the efficiency of AF detection consists of first selecting those individuals with a higher risk of AF, and the cheapest and quickest method to evaluate this risk is palpation of the pulse."

Response: Thank you for the suggestion, which we have followed (page 12).

6.3. "Indeed, systematic pulse assessment during routine clinic visits followed by 12-lead ECG in those with an irregular pulse has been already reported to substantially increase the detection of AF." That is correct, however this and many other studies have shown that opportunistic screening is likely to be even more effective and cost-effective than systematic approaches. This raises a question of whether the authors should have focused on creating an algorithm for pulse palpation in population groups for whom AF detection is already recommended, e.g. those aged 65 and over.

Response: The 2020 European Society of Cardiology AF guidelines recommend opportunistic screening for atrial fibrillation by pulse taking or ECG rhythm strip in those aged over 65 years. From our humility, we think that the paradigm of screening in patients over 65 years of age should be broken, because there are patients who are younger and have other risk factors, which we have included in our algorithm; they have a non-negligible risk of developing AF, so this group should also benefit from the effectiveness of a screening program. In this sense, we have included in the algorithm those most powerful factors related to the development of AF.

6.4. "In any case, our estimates may be not directly comparable to those of previous studies because most of latter did not clearly reported how they included or coded doubtful and undetectable palpations.10" Please change "reported" to "report".

Response: Thank you for noting the typo, which we have corrected (page 13).

6.5. "Morgan et al found also found high specificity (98%) but low sensitivity (54%) when considering an abnormal pulse as a 'continuous irregularity'.21" Remove one of the "found". Also, authors may wish to state the fact their results were perhaps closer to those of "frequent or continuous irregularity" defined by Morgan & Mant (sensitivity 72%, specificity 94%), which makes sense considering the criteria used by the present study for the positive test result.

Response: Thank you for noting the "found" typo, which we have corrected “Morgan et al. also found high specificity (98%) but low sensitivity (54%) when considering an abnormal pulse as a ‘continuous irregularity' (page 13). As suggested, we now also comment that " Our results were closer to the Morgan et al. definition of an abnormal pulse as a ‘frequent or continuous irregularity’ (sensitivity 72%, specificity 94%)" (page 13).

6.6. Reference 22. I could not relate this reference to data regarding the sensitivity and specificity of AF detection presented by authors. They may wish to review the reference selected accordingly.

Response: We have replaced reference 22: “Kavanagh S, Knapp M. The impact on general practitioners of the changing balance of care for elderly people living in institutions. BMJ. 1998;317(7154):322-327. doi:10.1136/bmj.317.7154.322” with the correct one “Sudlow M, Rodgers H, Kenny RA, Thomson R. Identification of patients with atrial fibrillation in general practice: a study of screening methods. BMJ. 1998;317(7154):327-328. doi:10.1136/bmj.317.7154.327”. 

“Sudlow2 et al. (page 13)

Thank you very much for the meticulous review and finding this mistake. We have carefully checked all the references, and found that reference 1 was also wrong, so it has been corrected.

6.7. "The importance of age in AF may has led many screening studies to use a threshold of ≥ 65 years.21,22" Please change "has" to "have".

Response: Thank you for noting the typo, which we have corrected. (Page 13)

6.8. "Younger people can also receive anticoagulation if they meet the CHADS-VASc score criteria, because 1-2 points of the criteria are age>65-75 years, but the other 7 points are not age-related.12" This statement is correct, however the low prevalence of comorbidities included in the CHA2DS2-VASc score (e.g. heart failure) amongst younger patients, means that the majority of < 65-year olds may not benefit from AF screening. Please change "CHADS-VASc" to "CHA2DS2-VASc".

Response: We now add that " However, the low prevalence of comorbidities included in the CHA2DS2-VASc score amongst younger individuals may mean that many may not benefit from AF screening " and have replaced "CHADS-VASc" with "CHA2DS2-VASc" (page 13).

6.9. "In the recent years, there have appeared new AF detection methods based on the use of novel technologies, such as smartphones with ECG electrodes, smart watches, or blood pressure machines." It is crucial not to combine single- or multiple-lead ECG devices and photoplethysmography-based devices/applications under the same umbrella: not just because the former may offer additional diagnostic potential for AF (30-second ECG is diagnostic of AF), but also because of differences in diagnostic accuracy (particularly specificity). I would urge caution when combining these technologies together considering the growing amount of evidence (including studies of cost-effectiveness) that supports the use of ECG-based devices in AF screening.

Response: Thank you very much for this appreciation. We have modified the text to give it more rigor as you have suggested.

“In the recent years, there have appeared new AF detection methods. These can be divided into record ECG tracings (single or multilead, in intermittent or continuous format, of various durations) and use non-ECG techniques such as pulse photoplethysmography, providing each method different diagnostic accuracy.” (Page 14)

6.10. "Conversely, our proposed screening would consist in a simple and inexpensive pulse palpation, which does not raise anxiety and is acceptable to most individuals.10" Suggest changing to "consist of".

Response: Thank you for noting the typo, which we have corrected “would consist of “(page 14).

6.11. "However, the sensitivity and specificity of a screening tool do not depend on the prevalence of the medical condition, and we checked that they neither depended on a range of heart conditions, so that we can safely translate the results to the general population." It is somewhat unusual that the presence of heart failure influenced the PPV/FOR but not the sensitivity or specificity. Am I correct in confirming this and what could be the underlying reason(s)?

Response: Even this is an interesting question, we have removed the first limitation, due to the arguments pointed out by the reviewer 2 (comment 9). Answering your observation, we checked that they neither depended on a range of heart conditions to safely translate the results to the general population. Here it is worth noting that while sensitivity and specificity do not depend on the prevalence, PPV and FOR are higher in subgroups with higher prevalence (e.g., patients with heart failure).

6.12. "However, the sensitivity and specificity of a screening tool do not depend on the prevalence of the medical condition, and we checked that they neither depended on a range of heart conditions, so that we can safely translate the results to the general population." As previously mentioned, despite the adjustments, the generalisation of these findings requires an adequate caution, considering they were drawn from cardiology patients and not patients in general practice.

Response: As we detailed in response to comment 1, we have added some sentences about the transferability of the results to general primary care.

6.13. Overall, the conclusion mostly draws on the findings of the study. On the other hand, at present I cannot support the use of term "primary care" instead of "outpatient" or "cardiology" clinic, taking into account the setting and nature of patients involved. Similarly, in the absence of substantial evidence, I would question the benefits of screening some patients aged < 65 years of age which is encouraged by authors of the study.

Response: As we specified in response to comment 1, to avoid confusion, we have replaced "primary care" with "cardiology outpatient clinics". Regarding the benefits of screening patients <65 years old, as we detail in response to comment 5.4, the paper now shows that palpation is not recommended for individuals <60 years unless they have heart failure. However, for 60-64 individuals, we would recommend palpation even if they do not have heart failure, given that the PPV is high (around 25%).

Finally in the results we said that there were 7860 pulse palpations. We have reviewed it and it was an error, they were 7844 , as stated in the abstract. We have corrected it.

---

## [Decision Letter · Decision Letter 1]

31 Mar 2022

Optimal opportunistic screening of atrial fibrillation using pulse palpation in cardiology outpatient clinics: who and how

PONE-D-21-32308R1

Dear Dr. Bañeras,

We’re pleased to inform you that your manuscript has been judged scientifically suitable for publication and will be formally accepted for publication once it meets all outstanding technical requirements.

Kind regards,

Giulio Francesco Romiti

Academic Editor

PLOS ONE

Additional Editor Comments (optional):

Reviewers' comments:

Reviewer's Responses to Questions

**Comments to the Author**

1. If the authors have adequately addressed your comments raised in a previous round of review and you feel that this manuscript is now acceptable for publication, you may indicate that here to bypass the “Comments to the Author” section, enter your conflict of interest statement in the “Confidential to Editor” section, and submit your "Accept" recommendation.

Reviewer #1: All comments have been addressed

Reviewer #2: All comments have been addressed

Reviewer #3: All comments have been addressed

Reviewer #4: All comments have been addressed

2. Is the manuscript technically sound, and do the data support the conclusions?

Reviewer #1: Yes

Reviewer #2: Yes

Reviewer #3: Yes

Reviewer #4: Yes

3. Has the statistical analysis been performed appropriately and rigorously? 

Reviewer #1: Yes

Reviewer #2: Yes

Reviewer #3: Yes

Reviewer #4: Yes

4. Have the authors made all data underlying the findings in their manuscript fully available?

Reviewer #1: Yes

Reviewer #2: Yes

Reviewer #3: Yes

Reviewer #4: Yes

5. Is the manuscript presented in an intelligible fashion and written in standard English?

Reviewer #1: Yes

Reviewer #2: Yes

Reviewer #3: Yes

Reviewer #4: Yes

6. Review Comments to the Author

Reviewer #1: This is a solid paper.

The authors have carried out an important study and presented the results clearly.

Reviewer #2: Authors responded to my questions and comments, and revised appropriately. I think revised manuscript is suitable for publication in PLOS ONE.

Reviewer #3: The authors have addressed or explained why they cannot address my recommendations and I am happy for the manuscript to be accepted in its revised form.

Reviewer #4: Authors have made substantial amendments to the original manuscript, particularly revising the title to reflect the study methodology and the structure of the manuscript itself in response to reviewers' comments. They have also addressed all questions submitted with the original review of the manuscript accordingly.

---

## [Editor Report · Acceptance letter]

12 Apr 2022

PONE-D-21-32308R1 

Optimal opportunistic screening of atrial fibrillation using pulse palpation in cardiology outpatient clinics: who and how. 

Dear Dr. Bañeras:

I'm pleased to inform you that your manuscript has been deemed suitable for publication in PLOS ONE. Congratulations! Your manuscript is now with our production department. 

Kind regards, 

on behalf of

Dr. Giulio Francesco Romiti 

Academic Editor

PLOS ONE